# Analysis of Inequalities and Inequities in Maternal Mortality in Chocó, Colombia

**DOI:** 10.3390/ijerph20126095

**Published:** 2023-06-09

**Authors:** Jorge Martín Rodríguez Hernández, Liany Katerine Ariza Ruiz, Daniella Castro Barbudo, Paula Vivas Sánchez, María Alexandra Matallana Gómez, Leidy Johanna Gómez Hernández, Lilibeth Romero Mendoza, Pablo Enrique Chaparro Narváez

**Affiliations:** Public Health Institute (PHI), Pontificia Universidad Javeriana, Bogotá 110231, Colombia; jrodriguez.h@javeriana.edu.co (J.M.R.H.); ariza.liany01@javeriana.edu.co (L.K.A.R.); daniella.castro@javeriana.edu.co (D.C.B.); mamata@javeriana.edu.co (M.A.M.G.); gomezh.leidyj@javeriana.edu.co (L.J.G.H.); lilibethromerom@javeriana.edu.co (L.R.M.); pchaparro@ins.gov.co (P.E.C.N.)

**Keywords:** maternal mortality, social determinants of health, maternal health services, Colombia, midwifery

## Abstract

We used a mixed design study to analyze the inequalities and inequities in Maternal Mortality (MM) for Chocó (Colombia) between 2010–2018. The quantitative component consisted of an analytical ecological design, where proportions, ratios, measures of central tendency and rates ratios, rate difference, Gini and concentration indices were calculated to measure inequalities. The qualitative component had a phenomenological and interpretive approach. One hundred thirty-one women died in Choco between 2010–2018. The Maternal Mortality Ratio was 224/100.000 live births. The Gini coefficient was 0.35, indicating inequality in the distribution of the number of MM with respect to live births. The health service offers have been concentrated in the private sector in urban areas (77%). The exercise of midwifery has played an important role in maternal and perinatal care processes, especially in territories where the State has been absent. Nevertheless, it occurs in complex circumstances such as the armed conflict, lack of transportation routes, and income deficits, affecting the timelines and care quality for these vulnerable groups. MM in Chocó has been a consequence of deficiencies in the health system and weaknesses in its infrastructure (absence of a high level of maternal-perinatal care). This is in addition to the territory’s geographical characteristics, which increase vulnerability and health risks for women and their newborns. In Colombia, as well as in other countries, many maternal and newborn deaths are preventable because their causes are due to social injustices.

## 1. Introduction

Maternal Mortality (MM) is defined as the death that occurs to a woman during pregnancy, childbirth or puerperium, regardless of its duration and place of occurrence, generated by any cause associated with or aggravated by gestation, the process of care and by non-accidental or incidental situations [1]. MM is generated by direct and indirect causes. The former includes abortion, hypertensive disorders, other pregnancy-related disorders such as bleeding or complications during childbirth and puerperium, as well as deaths due to obstetric sequelae; and the latter are related to pre-existing diseases that occur during pregnancy such as pyelonephritis, HIV-AIDS, among others [2].

MM is associated with three delays, correlated with the Social Determinants of Health (SDH). These are delay in seeking help: delay in going to a health center in a timely manner to assess signs and symptoms in the pregnant woman; delay in going to the health center: delay due to geographical aspects and socioeconomic difficulties that hinder timely travel; and delay in receiving adequate care: related to administrative barriers to receiving specialized, timely and efficient care [3].

The Maternal Mortality Ratio (MMR) is an indicator of MM, which reflects processes of inequality and inequity in health [4]. Contrary to inequalities, inequities in health are arbitrary, avoidable, and unfair differences that affect the health of people from different population groups and can be associated with characteristics such as socioeconomic income, social status, gender, discrimination, ethnicity, and lack of access to goods or services, among others. Inequities are usually assessed by means of SDH, using mainly quantitative strategies [5].

Goal 5 of the Millennium Development Goals stated that it was imperative to “improve maternal health,” which was estimated to reduce the MMR to 45 per 100,000 Live Births (LB). Colombia failed to meet this goal, although it did achieve a downward trend between 2000 and 2018. At the beginning of the millennium, MMR was estimated to be close to 105/100,000 LB; 2013 had the lowest incidence 58/100,000 LB, and in 2018, 80/100,000 LB [6]. In view of the failure to meet the Millennium Development Goals, new goals were reestablished for 2030, the Sustainable Development Goals, considering goal 3 to reduce MM to less than 70/100,000 LB.

MM also reflects the social, economic, geographic, administrative and cultural barriers and conditions of a Health System (HS). In other words, it is more accentuated in a territory where these SDH are combined. Administrative barriers occur in real access to health care, especially with regard to seeking and continuing treatment. These result from problems related to infrastructure, professional resources and delays in authorizations [7].

With regard to social and economic determinants, it has been estimated that in upper-middle-income countries such as Colombia, most women who died during or after gestation had conditions of poverty, reflected in inadequate housing and utilities, critically overcrowded households, high economic dependence and school-age children who did not attend school [8,9,10]. Inequalities in MM are related to conditions of territorial poverty, [3] aspects of HS linked to coverage and governance, [9] low levels of schooling and inadequate provision of policies that guarantee care for vulnerable populations [10].

In this context, it is essential that the service offered and the installed capacity be oriented to face the region’s demographic, epidemiological and social reality. Therefore, health services that are close and accessible to vulnerable and distant populations should be guaranteed [11]. It is important to know the availability of gynaecobstetric services and transport assistance, delivery rooms, and obstetric beds in the municipalities for the safe care of women. It is also important to know the dynamics associated with community response, an essential aspect of maternal health care in ethnic and rural communities in the country.

This article aims to identify and analyze inequalities and inequities in MM for the department of Chocó between 2010–2018. We explore the hypothesis if most of the MM are potentially avoidable and if they are related to conditions of social marginality, Unsatisfied Basic Needs (UBN), low employability, scarce access to roads to populated centers, precariousness in the care network, discrimination and exclusion, among other conditions that evidence the inequity among pregnant women in this territory. This is subject to global structural determinants, which place poor women living in rural areas and ethnic minorities in a situation of social disadvantage [12].

## 2. Materials and Methods

### 2.1. Design and Study Population

This research corresponds to a mixed design. The quantitative component was developed with an ecological-analytical design, using secondary information sources disaggregated by the municipality to describe and analyze the behavior of MM, MMR and some sociodemographic indicators potentially associated with MM, as well as information on the institutional supply of maternal services.

The qualitative component developed a phenomenological and interpretative approach. In order to understand health inequalities, experiences and perceptions of community and institutional actors on maternal care and aspects related to MM were investigated.

### 2.2. Area of Study

Chocó is located in the Colombian Pacific and comprises 30 municipalities. It has 46,530 km^2^, 90% of which are special conservation areas. The majority of the population is black or Afro-Colombian (69.1%), followed by indigenous (12.8%), raizal (0.01%), palenquera (0.02%) and ROM (0.01%). More than half of the people live in rural areas (55.4%).

### 2.3. Quantitative Aspects

#### 2.3.1. Source of Data

We used data from the Ministry of Health and Social Protection, such as the Integrated Social Protection Information System for maternal care statistics; records from the National Administrative Department of Statistics from 2010 to 2018 to identify information on MM and LB, and the Special Registry of Health Service Providers to establish the institutional offer and installed capacity. In addition, the National Administrative Department of Statistics 2018 Census was used to describe the population characteristics of each municipality in Choco. The UBN Index and the municipal Multidimensional Poverty Index (MPI) were used to identify the socioeconomic level.

#### 2.3.2. Study Variables and Data Processing

The variables used were: year of death, number of MM per year, number of LB per year, UBN, MPI, the proportion of affiliation coverage to the social security system, childbirths attended by a specialized professional, institutional childbirths, LB with four or more prenatal care controls, afro and indigenous populations, index of obstetric beds, public and private Health Care Facilities Institutions—HCFI (IPS—Instituto Prestador de Servicio de Salud) by the municipality, services enabled for maternal health according to the level of complexity by the municipality, enabled assistance transport services, delivery rooms and obstetrics and gynecology services. MM, LB and MPI were used to estimate inequality measures. Health care services reported by national legislation were included.

### 2.4. Data Analysis

Proportions and ratios were calculated for categorical variables, average, medians (by non-normal distribution), standard deviation, coefficient of variation, maximums, minimums, and interquartile range were used for quantitative variables. The departmental and municipal MMR was calculated. The measurement of inequalities was calculated with MMR, rate ratio, rate difference, absolute and percentage Population Attributable Risks (PAR and PAR%), Gini index and concentration index. In addition, Lorenz and Concentration curves by the municipality were constructed. Microsoft Excel, Stata 14 and EPIDAT 4.2 were used.

### 2.5. Qualitative Aspects

The qualitative aspect of this study aims to understand the context of the community, observe the proffer of institutional care and attention, and identify the main health problems of pregnant women in the department of Choco-Colombia.

For this purpose, a series of semi-directed interviews were designed and applied at the community level. These interviews focused on various issues such as:(i)Midwives’ trajectory in care and attention to pregnant women and to newborns.(ii)Knowledge and practice of traditional medicine.(iii)Relationship between medicines (allopathic care and traditional care), and(iv)Complications management during pregnancy, childbirth and postpartum.

Besides the institutional level, the topics discussed were:(i)Organization and operation of health care networks and institutional providers;(ii)Barriers to access to health services;(iii)Maternal-perinatal care response;(iv)Adequacy of the differential approach

In 2019, 20 interviews were conducted in the Chocó department. From November 1 to 4 in the community of Yuto (Choco), during the meeting of Choco’s Midwives, 16 interviews were conducted with Asoredipar-Choco directors, leaders and midwives. The selection criteria of the interviewees were the knowledge of the work as a midwife in Chocó and the Pacific coast.

From November 5 to 7 in the city of Quibdó (Chocó), different visits were made to officials of Health Provider Institutions and Territorial Health Directorate, organizations with actions in maternal and neonatal health issues. These visits were arranged between the research team and the local health authorities. Just 4 people were interviewed.

### 2.6. Data Processing and Analysis

The interviews lasted approximately 1 h and 30 min; all were recorded and transcribed verbatim, then a coding exercise with deductive and inductive categories was carried out using the software NVivo, version 12.

For the analysis of the primary information, open and axial coding techniques were used. At the end of this process, an initial description of findings by categories was made. These aspects were constantly discussed, analyzed, and triangulated by the researchers. This process allows the researchers to redefine and reorganize new categories and identify how all the categories correlate. Subsequently, the final analysis was accomplished, including the report’s synthesis, interpretation, and writing.

## 3. Results

The quantitative findings reveal that the MMR showed irregular behavior between 2010–2018, ranging from 180 to 377 maternal deaths per 100,000 LB. Five municipalities did not report MM: Juradó, Carmen de Atrato, Cantón de San Pablo, Unión Panamericana and Carmen de Darién. Seven municipalities presented MMR greater than 500 deaths per 100,000 LB: Bajo y Medio Baudó, Bojayá, Litoral de San Juan, Novita, Rio Iro and Río Quito.

Between 2010 and 2018, there were 131 deaths of pregnant women (49 in Quibdó) and 58,352 LB (29,091 in Quibdó and 252 in Litoral de San Juan). The MMR in Chocó was 224/100,000 LB, the highest in Litoral de San Juan (1190). The minimum MPI was in Quibdó (44.4) and the maximum in Alto Baudó (90.6). Carmen de Atrato recorded the highest percentage of afro-descendant population and Atrato the lowest. Bagadó had the highest percentage of the indigenous population and Condoto the lowest. The lowest General System of Social Security in Health affiliation coverage was in Medio Atrato (26.4%). Medio San Juan did not report institutional childbirth coverage or care by specialized personnel. Unión Panamericana reported the highest percentage of Prenatal Care (85.7%). Carmen de Darién had the lowest Obstetric Beds Index (0.2) and Sipí the highest (2.5).

Regarding MM inequality indicators, 3.1 times more pregnant women died in Alto Baudó (with the worst MPI) than in Quibdó (with better MPI). The difference between MMR of these two municipalities was 354.4 MM/100,000 LB. According to the PAR%, if all municipalities in Chocó had had the MMR of Quibdó, deaths would have been reduced by 25.1% (Table 1).

The Gini coefficient was 0.35, indicating inequality in the distribution of the number of MM in relation to LB. The Lorenz curve showed that 30% of the MMs occurred in municipalities that contributed 10% of the LBs (Figure 1). The MMR concentration index according to MPI was −0.19, where the municipalities with the worst MPI accumulated more MM than expected. Forty percent of the MMs occurred in municipalities that contributed 25% of the LB and had higher MPI (Figure 1).

### Institutional and Community Response

For pregnancy, childbirth and postpartum care, the community response is identified by midwives, traditional doctors and community leaders. The institutional response involves state authorities such as the Ministry of Health, Departmental and Municipal Health Secretariats, Benefit Plan Management Companies (BPMC), and private and public HCFs. The offer of services is concentrated in private HCFs in the urban area (77%) (Table 2 and Figure 2).

Health services show that 88% of maternal and neonatal care corresponds to low complexity, 9.3% to medium complexity, 2.4% to low-medium complexity, 0.5% to medium-high complexity, and 0.1% to high complexity (in Quibdó and Istmina). The highest proportion of gynaecobstetric services (1.8%) is offered by the private sector in Acandí, Istmina, Quibdó and Tadó, while the public sector offers (0.12%) in Quibdó and Tadó. In 2019, 35 delivery rooms were reported in 24 municipalities, except Bahía Solano, Juradó, Medio Atrato, Sipí, Unguía and Rio Quito.

From the qualitative findings, we found that in terms of transportation, Chocó has a maritime ambulance in Acandí and Armed Forces support for air transport in very remote areas in cases of vital emergencies. The administrative procedures depend on the BPMC and may result in referrals to other municipalities outside the department. Access to health services is also mediated by geographic and social conditions in the department; for example, pregnant women and their newborns travel on paths and/or roads in poor condition, on foot, on animals or motorcycles, in canoes or boats on long river journeys. Mobilization also depends on confrontations between armed actors, weather conditions such as torrential rains, and resources to cover transportation, food and lodging.

Afro-Colombian and indigenous midwives, representatives of health care in their communities, are found in scenarios with scarce and precarious institutional offerings. The community response is based on a belief system that integrates natural and supernatural aspects, as well as biomedical knowledge and practices. This response also promotes identity, political processes and the safeguarding of their traditions. In 2019, Asoredipar-Chocó identified 880 collaborators (732 female midwives, 63 male midwives and 85 without clear gender), 488 Afro-Colombians, 238 indigenous Embera (Chami, Dobida and Katío), 30 mestizos and 124 without clear ethnicity. In the last 10 years, it is estimated that they have attended nearly 18,000 births in the department.

Despite their important role, midwifery is carried out amid a complex reality. Some have been victims of multiple violence/abuse in their territories, such as the armed conflict, in addition to the lack of roads, transportation, opportunities and income, which affects the timeliness and quality of care:


*“A midwife is very important for many reasons, imagine, you know that the Secretary of Health doesn’t cover all the townships and villages that are many in our municipality, no matter how hard it tries. From here down there are townships from where it takes six hours to reach the urban area, and If a midwife is well trained, in an emergency she could save many lives for both mother and baby” Interview Istmina midwife.*


Their care does not always arrive where and when needed, nor do they have the necessary supplies for said care. In this regard, several midwives reported needing items to attend births, such as gloves, blades, gauze, and flashlights, among other things. They report having received training from health organizations and entities, in addition to receiving supplies for care. However, these are not always replenished, so they end up attending with what they have at hand.

In some areas of the department, the number of midwives has also decreased, and some do not want to continue with this profession because they have been constantly delegitimized, discriminated against, or because their religious beliefs prevent them from exercising this role:

In the words of a midwife […] *there are some who are very smug because they work in the hospital, there are others who don’t let you in, they see you as nothing. They don’t let us in, others who say that without uniforms they don’t enter or ‘me on my shift I do not accept midwives’, others say that we are chinangas* [witches]. Interview Condoto midwife.

In some communities, the transmission of knowledge to younger generations is not taking place either due to the lack of interest in this work. The result is that women and their children face, in some territories, an almost total absence of community and institutional care. In this context, Asoredipar-Chocó has promoted spaces for dialogue, exchange, and knowledge transfer between medicines for more than ten years, seeking to strengthen the territorial community response.

Physical and cultural distance from the HCF-Indigenous was identified, which are perceived as places that over-medicalize the body, do not provide respectful treatment, and perform procedures without consent. As a result, in some cases, women choose not to go to these institutions and prefer to seek care from midwives. This “tension” between biomedicine and traditional medicines also appears in the relationship between the actors of these medicines. In this regard, it was found that the community response encourages women to go to health institutions, reiterating the importance of prenatal check-ups and examinations, micronutrient intake, assessment, and the beginning of the vaccination schedule for newborns. Despite this approach, midwives express that in health institutions, they are rejected and accused of putting the life of the woman and the baby at risk, designating their work in the margins of informality or that they do not comply with the standards of care, generating resistance to not interact with health professionals.

Childbirth turns out to be a critical moment because the greatest encounters and misunderstandings occur between the actors of medicine. On some occasions (the minority), they seem to be allies, with reciprocal recognition and joint attention. Still, on other occasions, there is finger-pointing and mistreatment of the midwives that can delay the search for help, even when the situation requires it.

What has been presented up to this point makes visible the difficult conditions faced by pregnant women in Chocó, who, exercising one of the most important social roles: giving life to another human being, put their lives and those of their children at risk. This difficult situation was summarized by a participant as follows:


*“ Anyway, I don’t know, I’d think that in the midst of everything, it’s surprising that pregnant women don’t die on us, see, let’s say that some do and you get worried, but given the outlook as you can see, given so many difficulties, I’d think that it’s a small number, small because there are always many difficulties for the care of both pregnant women and newborns...”.*
Interview Territorial Health Entity, 2019

## 4. Discussion

The behavior of MMR between 2010–2018 was irregular, with significant differences between municipalities. Through some demographic, socioeconomic and health indicators, it was shown that inequalities and inequities are related to MM in the department of Chocó.

Latin America is one of the regions with the greatest income inequality, making it one of the most inequitable areas in the world. According to reports by the Economic Commission for Latin America, Colombia is among the countries with the greatest social inequality, reflected in one of the highest Gini coefficients (like Brazil and Panama) [13]. Within the country, there are geographic areas with great social inequalities; several municipalities in La Guajira, Chocó, Amazonas, Vaupés and Guainía have MPIs above 80% [14].

Chocó is a remote and disconnected territory. Its national Gross Domestic Product share is 0.39% [14], its UBN is 89.5%, poverty is 29.3%, overcrowding is 22.8%, schooling is low and its Gini is 0.56. Throughout history, it has experienced the absence, weakness and corruption of the State, in addition to the interests of illegal groups with territorial disputes over mining areas, illicit crops and drug trafficking routes [15]. Most of its mobility is fluvial or maritime, depending on access to gasoline, engines for boats and the navigability of its rivers. By land, it communicates with other national places and cities. However, the poor condition of its roads makes mobility difficult. It has the lowest level of paved kilometers in all of Colombia. This situation has led to the isolation of rural areas from their capital and the rest of the country. It has four airports, but only the one in Quibdó functions adequately [10].

In Chocó, there are large gaps in the risk of MM among its municipalities. The most affected were those with high proportions of the population with MPI. This indicator reflects poor educational conditions of the household, children and youth, health, work, housing and access to public utilities. This situation is similar to that described at the beginning of this century, when high inequality was observed when analyzing the MMR compared to the UBN [8]. These circumstances result from the poor social and economic conditions of each of the municipalities and the deficient quality of the HS [16].

The Colombian HS’s design and its insurance dynamics place the BPMC in a privileged position, where their decisions become structural determinants for the care or not, of women and their children. The institutional offer in Chocó shows unequal conditions; for example, the private sector concentrates its offer in urban areas, making the availability of services in rural areas precarious. In addition, the care resolutive capacity is limited by insufficient infrastructure, equipment and human talent in low and medium complexity services. For some authors, if the Health System is consistent with guaranteeing access to health care for all, it should not create obstacles to providing the required care [17].

The centralization of health services in urban areas is not a situation that has happened only in Colombian territories. Other studies carried out in Mexico, Peru, Ecuador and Guatemala had detected that women living in rural areas have greater barriers to prenatal care, childbirth, and the puerperium. This is associated with the poor presence of institutions in those areas and the low accessibility of health services for women who live far from urban centers. Moreover, afro-descendant and indigenous women have experienced inequalities in the health system because it is anchored to a mestizo society that does not know the local language, cultural practices and beliefs [18,19,20].

PAHO reports that social determinants can exert a powerful influence on access and quality of care [21]. One of the purposes of the HS is to minimize inequities and inequalities through equitable access to health services [22]. However, as observed in Chocó, the Health System behaves as a structural determinant that increases its inhabitants’ social, economic and human costs by not responding to their felt needs, in our case, for pregnant women and newborns.

What has been described above coincides with the concept of bureaucratic itineraries and indicates that the Colombian Health System functions as a structural barrier to access to health services generating health inequities for those populations [23]. Other studies worldwide have found that bureaucratic processes act as barriers to accessing maternal services and influence the choice of women from private health institutions for the birth of their children in which the access to care is less [24,25].

Some researchers consider that MM may be the result of a poorly accessible and culturally inadequate HS, associated with a chain of structural factors such as social position, food security, schooling, access to and quality of health services that act individually and collectively. This contributes to women’s disadvantaged position during pregnancy, childbirth and postpartum, increasing risks and complications [26]. The above are clear examples of health inequities for Latin-American and African territories, where there are unfair and avoidable situations [22] that reflect the lack of coherence in many areas of health policies for society [27].

In the communities studied, women prefer midwifery care because of the social and cultural proximity amid a context with low state presence and poor quality of health services. This agrees with other authors, who have identified that cultural and geographic proximity mediates women’s choice of where and how their children are born [18,28,29].

This research suggests that power relations hinder the relationship between medicines and their actors, anchored in the hegemony of biomedical knowledge considered as unique, legitimate and effective. Community medical systems such as those of Chocó are located in symbolic and social margins [30], usually associated with risky practices and obstacles to the populations’ health and welfare development [31]. This agrees with other studies, where power-knowledge relations hinder the relationship between medicines [32], generating epistemicides of other forms of knowledge [33,34,35].

These difficulties materialize in the treatment received by women in the HCF-I, whether they are pregnant women or midwives, due to the low incorporation of intercultural approaches in health care, which can lead to delays in seeking care. Some authors point out that the lack of intercultural adaptation leads to communication and quality of care problems and accentuates inequalities with personnel who do not know the cultural codes, affecting the rights of pregnant women who do not receive sufficient, timely and quality information and care [18]. Other studies have identified that treatment and care make ethnic communities seek health institutions when the disease processes are irreversible [31,32,33,36,37]. The results of this study also point to difficulties in the generational transmission of midwifery. In contexts lacking HSPIs and where the existing ones do not respond to the population’s needs, the disappearance of this community resource would leave women and their children to their fate, increasing the possibilities of MM.

That study has some limitations. For example, among the quantitative limitations, it should be noted that mortality information coming from secondary sources may have quality errors in the registry, and measurement errors, among others. In Chocó (2002), the estimated coverage of deaths was 64.5%, and the masking of the cause in the measurement of MM was 24% (non-coverage) [38]. Additionally, the disaggregate information of the municipalities does not show the extent and social differences of these municipalities. The qualitative limitations are related to the generalization of the results, as these mainly represent the interviewees’ perspectives, so other people in the territory may disagree with the ideas expressed in this research.

The MM in Chocó, as in other parts of the world, is a consequence of the cultural, economic and political SDH [39], associated with power constructions on ethnic and racial differences, unequal access to opportunities, and rights [27,37,39,40]. Derived from intersectionality, inequalities are accentuated by gender relations, causing MM to be higher in indigenous women who live in rural areas and who have no resources [12].

Moreover, worldwide, other sociodemographic determinants are potentially associated with the incidence of maternal and perinatal morbidity and mortality. For example, some micronutrient deficiencies such as thiamine, folic acid, magnesium, Vitamin B12, and others could be associated with these issues. These nutritional deficiencies may produce health problems like Wernicke’s encephalopathy, hypoxic-Ischemic encephalopathy neonatal, neural tube defects, oral clefts, neural tube defects, limb defects, and urinary tract anomalies, among others [41,42,43,44].

Many of these micronutrients are included in several foods, such as meat, fish, poultry and vegetables [45]. As was previously mentioned, the Choco population are poor people with low levels of income, and maybe they do not have enough incomes to access these sources of meals. In this research, we could not establish these topics deeply.

Although acknowledging cultural differences in health helps improve the relationship and community response in the territories, this is not enough to reduce MM and infant deaths, which require structural changes in the living conditions of pregnant women. In this sense, reducing these deaths implies an intersectoral and interinstitutional commitment that guarantees the achievement of wellbeing and health as a fundamental and constitutional right. Some peripheral communities of Colombia, as in other places of the world, need to implement a Universal Health System to ensure access to care, regardless of pay ability, as has been recommended by PAHO [21]. As we can see in other contexts, market systems, such as those operating in Chocó, increase social health inequities in those territories [46].

## 5. Conclusions

According to the findings of this study, In Colombia several regions of the country are isolated and disconnected due to development issues. The Chocó department is an example, due to the regional differences the population lives under unequal conditions. These complex conditions that impact the health of the population require new and more investments because it is not possible to apply the same health model for this population as in the central regions of the country.

It is necessary to adopt a new approach in these complex areas, with the purpose to reduce the social health inequities and inequalities among the Choco population, developing activities in favor of the population not only implies focusing on women and children, also it is important to launch a large investment that involves actions in different areas as: health, nutrition, education, employment, and mobility among many other areas that were identified in this study. Moreover, in these territories, it is necessary to improve the skills of midwives and give them tools and elements to develop the midwifery exercise. Tools such as communication systems that allows them to receive guidance, to report some topics associated with their patients, to request economic resources for the patient´s transport, and to request the elements to attend deliveries, among others.

In addition, we must not forget that the transformation of inequities and inequalities in health that affect pregnant women and newborns in Chocó, and that are presented in this article, requires a transformation from the structures on which they are based, such as the systematic racism that leads to the invisibility and normalization of the multiple discriminations and social exclusions of Afro-Colombians people and other ethnic groups.

## Figures and Tables

**Figure 1 ijerph-20-06095-f001:**
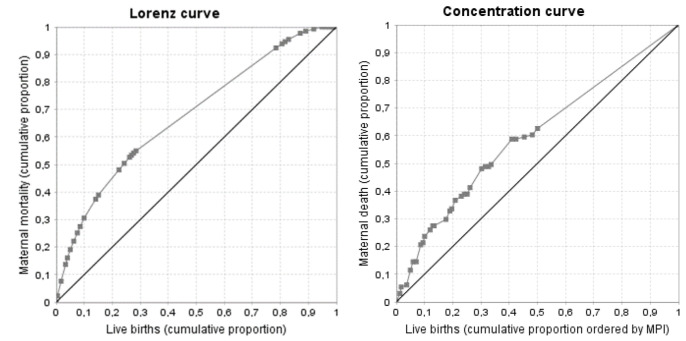
Lorenz and concentration curves according to MPI in Chocó. 2010–2018.

**Figure 2 ijerph-20-06095-f002:**
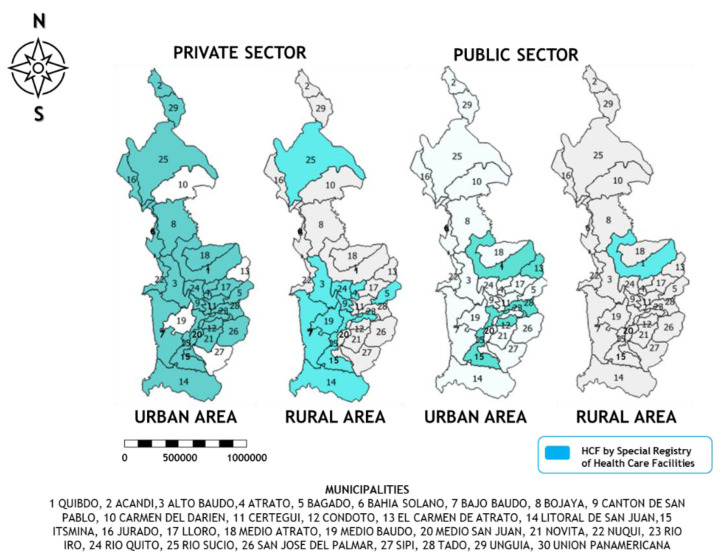
Maternal-neonatal HCF by legal status is reported in the Special Registry of Health Care Providers.

**Table 1 ijerph-20-06095-t001:** Social and demographic indicators. Maternal Mortality, Chocó 2010–2018.

Indicator	Median	Average	SD	IQR	Min	Max	CV
Maternal deaths (*n*: 131)	2.5	4.4	8.9	3	0	49	202.3
Live births (*n*: 58,352)	750.5	1.942	5.193	463	252	29.091	267.4
Maternal Mortality Ratio (MMR)	242.1	323.9	311.7	431	0	1.190	96.2
UBN	55.5	56.8	15.3	21	24	83.9	26.9
MPI	68.3	66.7	11.8	19	44	90.6	17.7
% affiliate coverage SGSSS	88.1	80.5	21.1	37	26	100	26.2
% institutional delivery coverage	86.5	74.6	27.7	30	0	100	37.1
% delivery attended by specialized personnel	85.6	75.7	27.6	33	0	100	36.5
% live births with four or more prenatal care visits	51.5	51.3	20.3	25	0	85.7	39.6
% Afro-descendant population	76.8	67.8	24.3	40	4	96.1	35.8
% Indigenous population	11	18.4	18	23	0.6	58.7	97.8
IOB per 1000 WCA (15–49 years)	0.8	0.94	0.51	0.7	0.2	2.47	54.3

SD: Standard deviation; IQR: Interquartile Range; Min: Minimum; Max: Maximum; CV: Coefficient of variation; UBN: Unsatisfied Basic Needs; MPI: Multidimensional Poverty Index; IOB: Index of Obstetric Beds; WCA: Women of Childbearing Age.

**Table 2 ijerph-20-06095-t002:** Proportion of health care facilities institutions reported in Registry of Health Minister.

Geographic Area	Legal Status	Hcf (N)	%
Rural area	Private sector	10	14%
	Public sector	1	1%
Rural area Total		11	15%
Urban area	Private sector	64	77%
	Public sector	6	7%
Urban area Total		70	85%
Total		81	100%

## Data Availability

Not applicable.

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
