# Peer review of "Analysis of Inequalities and Inequities in Maternal Mortality in Chocó, Colombia"

_ijerph, 2023, doi:10.3390/ijerph20126095_

Round 1

Reviewer 1 Report

Thank you for the opportunity to review this paper.
The topic is interesting and relevant for both readers and cultural context. I think the manuscript has potential to be published, but it needs for adjustments to be improved. Here below I suggest some recommendations.

1) Methods: this section is missing a lot of information. The authors should better describe the method used for qualitative investigation.

- how was the selection of the 24 participants done?
- how were they recruited and contacted?
- how many were recruited?
- among those recruited, did any refuse to participate?
- How and where were the interviews conducted?
- what was the interview duration (mean and range)?    

2) In "results" section, the authors should specify which findings are derived from the quantitative survey and those from the qualitative survey. For the qualitative part, the authors should also report an interview excerpt for each topic addressed. what were the characteristics of the 24 participants? missing demographics.

3) Line 304, I suggest the authors to change "conclusions" to "discussion".
English should a bit improved. For example, I suggest to change "Source of data used" to "Source of data" (Line 110).

Author Response

Dear reviewer:

Thank you very much for reviewing our article.

The authors have made the adjustments and modifications that you can see in the new version that we are attaching.

  1. Methods: this section is missing a lot of information. The authors should better describe the method used for qualitative investigation.
  • How was the selection of the 24 participants done?
  • How were they recruited and contacted?
  • How many were recruited?
  • Among those recruited, did any refuse to participate?
  • How and where were the interviews conducted?
  • What was the interview duration (mean and range)?

 Adjustment: The changes were made between lines 139 and 174, in these lines the description about the qualitative methods was added as the evaluator suggested.

  1. In the "results" section, the authors should specify which findings are derived from the quantitative survey and those from the qualitative survey. For the qualitative part, the authors should also report an interview excerpt for each topic addressed. what were the characteristics of the 24 participants? missing demographics.

Adjustment: Quantitative results can be identified on line 173 and qualitative findings on line 261. The narratives were prioritized according to their relevance to the topic and according to the authors, these narratives support the findings presented in the manuscript, the narratives were included from lines 271 to 279 and from lines 286 to 293, considering that by including this information the maximum number of words allowed for the article is not compromised.

  1. Line 304, I suggest the authors to change "conclusions" to "discussion". English should be a bit improved. For example, I suggest to change "Source of data used" to "Source of data" (Line 110).

Adjustment: All the suggested changes have been made, in line 110 Source of data. On line 328 discussion was left.

Reviewer 2 Report

it is suggested that in the results of the paper, some relevant comments of the semi-structured interviews carried out should be added

Author Response

Dear reviewer:

Thank you very much for reviewing our article.

The authors have made the adjustments and modifications that you can see in the new version that we are attaching.

  1. It is suggested that in the results of the paper, some relevant comments of the semi-structured interviews carried out should be added.

Adjustment: In the new version some narratives are included in lines 277-281 and 292-295, considering that including this information the maximum number of words allowed for the article is not compromised.

Reviewer 3 Report

In a quantitative and qualitative ecological analysis, Jorge Martin Rodriguez Hernandez and colleagues attempt to identify and analyze inequalities and inequities in maternal mortality (MM) in the department of Chocò , Colombia, between 2010-2018. The topic is of considerable importance in social and clinical settings. The manuscript is interesting and well-analyzed  and my comments are made in a attempt to be of help for improving its worth.

Considered that among the many sociodemographic determinants of health potentially associated with the occurrence of maternal mortality in Colombia, deficiency of micronutrients such as thiamine, folic acid, magnesium and others (Wedisingle L. et al., Wernicke's encephalopathy: a preventable cause of maternal death. Br J Hospital Med. 2011; 72:1; Got YI, Koren G. Folic acid in pregnancy and fetal outcomes. J Obstetrics Gyneacology 2008; 28:1; Sechi G et al., Thiamine as a possible neuroprotective strategy in neonatal hypoxic-ischemic encephalopathy. Antioxidants (Basel) 2021; 11:42) may play an important role. I suggest to further discuss this item and to quote in the manuscript the references indicated.

Minor points:

-Line 130. "Average" was not listed among the mean values, yet it appeared in Table 1 on page 4. CV is obviously a %CV.
-Page 4, table 1. In row 3 (live births) the numbers represent thousands, so they should contain the comma and not the decimal point. Please check.
-Line 275. "Asorepidar" was mispelled, correct to Asoredipar Chocó as in line 255.
-Line 353. Please add the subject: "as it has been observed"

Author Response

Dear reviewer, 

Thank yoy very much for reviewing our article. 

The authors have made the adjustments and modifications that you can see in  the new version that we are attaching.

  1. I suggest to further discuss this item and to quote in the manuscript the references indicated.

Adjustment: The references recommended by the reviewer have been included in the discussion section and were related to the findings mentioned in the manuscript. See lines 433 to 443. These references and others can be verified in lines 593 to 605.

  1. Line 130. "Average" was not listed among the mean values, yet it appeared in Table 1 on page 4. CV is obviously a %CV.
    Page 4, table 1. In row 3 (live births) the numbers represent thousands, so they should contain the comma and not the decimal point. Please check. -Line 275. "Asorepidar" was mispelled, correct to Asoredipar Chocó as in line 255.

Line 353. Please add the subject: "as it has been observed"

Adjustment: The changes recommended by the evaluator in lines 130, 131 and table 1 have been made. Moreover, in lines 183-184 we made changes ICO and MEF by IOB [Index of Obstetric Beds] and WCA [Women of Childbearing Age]. The word Asoredipar was adjusted in line 299 and the subject "it" was included in line 377. 
